# The presence of spin in systematic reviews focused on diabetic neuropathy: A cross-sectional analysis

Ali Khan[1]*, Haley Riley[1], Ryan Ottwell[1,2], Wade Arthur[1], Benjamin Greiner[3], Ekaterina Shapiro[4], Drew Wright[5], Micah Hartwell[1,6], Suhao Chen[7], Zhuqi Miao[8], Stacy Chronister[4], Matt Vassar[1,6]

1 Office of Medical Student Research, Oklahoma State University Center for Health Sciences, Tulsa, Oklahoma, United States of America, 2 Department of Internal Medicine, University of Oklahoma, School of Community Medicine, Tulsa, OK, United States of America, 3 Department of Internal Medicine, University of Texas Medical Branch, Galveston, Texas, United States of America, 4 Department of Internal Medicine, Oklahoma State University Medical Center, Tulsa, Oklahoma, United States of America, 5 Samuel J. Wood Library & C.V. Starr Biomedical Information Center, Weill Cornell Medical College, New York, New York, United States of America, 6 Department of Psychiatry and Behavioral Sciences, Oklahoma State University Center for Health Sciences, Tulsa, Oklahoma, United States of America, 7 School of Industrial Engineering and Management, Oklahoma State University, Stillwater, Oklahoma, United States of America, 8 School of Business, State University of New York at New Paltz, New Paltz, NY

* ali.s.khan@outlook.com

**Data Availability Statement:** All relevant data are within the paper. Additionally, all raw data forms

## Abstract

### Background

Spin—the misrepresentation of a study's actual results—has the potential to alter a clinician's interpretation of the study's findings and therefore could affect patient care. Studies have shown spin frequently occurs in abstracts of systematic reviews from a variety of other medical disorders and specialties.

### Aims

Our primary aim was to evaluate whether the nine most severe types of spin occurred in systematic review abstracts' concerning diabetic neuropathy treatments. Secondly, we aimed to determine whether spin presence was associated with the methodological quality of a systematic review.

### Methods

A search of MEDLINE and Embase collected 1297 articles focused on diabetic neuropathy treatments, of which we included 114 systematic reviews for spin assessment. Each included study was evaluated for the nine most severe types of spin as defined by Yachitz et al. The methodological quality of a systematic review was determined by using the AMSTAR-2 instrument. All screening and data extraction were conducted in a masked, duplicate fashion. Since the final sample size of 114 was not sufficiently powered to do multivariable logistic regression, we calculated unadjusted odds ratios which evaluated relationships between spin presence within abstracts and study characteristics.

from the study can be found in the open science frame link (https://osf.io/7e5nd/).

**Funding:** MV received the award. The development of our protocol was funded by the Oklahoma State University Center for Health Sciences Presidential Mentor-Mentee Research Fellowship Grant. The funders had no role in study design, data collection and analysis, decision to publish, or preparation of the manuscript.

**Competing interests:** The authors have declared that no competing interests exist.

## Results

From the 114 articles reviewed, spin was present in 7.9% of the studies (9/114), with spin type 5: "conclusion claims the beneficial effect of the experimental treatment despite the high risk of bias in the included primary studies" as the most frequent in our study. Spin types 1, 2, 6, and 8 were not identified. No association was observed between the study characteristics and spin presence, including the methodological quality of a systematic review.

## Conclusions

Overall, spin is infrequently observed in abstracts of systematic reviews covering diabetic neuropathy treatments. When comparing our results to other fields of medicine, the field of diabetic neuropathy research publishes systematic reviews whose abstracts mostly portray the findings of the review's full-text to reflect the results adequately.

## Introduction

Nearly 10% of the United States adult population is living with diagnosed diabetes (either type 1 or 2), with an additional 3% estimated to be living with undiagnosed diabetes [1]. Among those living with diabetes, diabetic neuropathy is one of the most common clinical complications, present in over 30% of these individuals [2,3]. Apart from strict glucose control, few therapeutic agents are FDA approved to control the symptoms of diabetic neuropathy [3]. In order to provide the appropriate care to patients with diabetic neuropathy, physicians must stay updated through research and various scientific literature. Systematic reviews and meta-analyses serve as an excellent method for physicians to keep their medical knowledge current.

Systematic reviews are widely recognized as the gold standard for a high level of research evidence and typically represent the most thorough summaries of current literature [4]. Systematic reviews use explicit systematic methods to reduce the risk of bias in the review process, therefore they are considered the highest level of evidence [5]. Given this high standing, they are supposed to report findings free of spin. Yachitz et al. classify spin based on three overarching categories: misleading reporting, misleading interpretation, and inappropriate extrapolation [6]. While the entirety of a scientific work should maintain a high professional standard, there is one section that should be more thoroughly scrutinized—the abstract. As the portion most commonly read by clinicians, the abstract has the greatest potential to alter the course of patient treatment and outcomes [7].

Spin has been identified as a broadly used form of bias in randomized controlled trials within various fields, as well as, specific disease states [8,9]. However, discrepancies have been found between specialties. For example, 70% of otolaryngology abstracts contain spin in comparison to 44.3% of emergency medicine abstracts, and 27.3% of cardiology abstracts [9–11]. Due to the variability in results of past research, further exploration is warranted to examine the presence of spin in additional fields. One unexplored topic is diabetic neuropathy and the presence of spin within treatment options. As diabetic neuropathy is highly prevalent in the US, the evaluation and maintenance of scientific integrity are vital.

The purpose of this study was to analyze the nine most severe types of spin as defined by Yachitz et al. [6] within abstracts of systematic reviews focused on diabetic neuropathy treatment options. Our secondary objective was to determine whether spin was associated with particular characteristics of systematic reviews, including their methodologic quality.

## Methods

### Protocol and reporting

To ensure our study's reproducibility and transparency, this study's protocol and data analysis scripts can be found in Open Science Framework (https://osf.io/7e5nd/) [12]. Additional studies examining the presence of spin in systematic reviews in different medical fields were conducted simultaneously with this study. Additional statisticians repeated analysis to further ensure the reproducibility of our work. We used Preferred Reporting Items for Systematic Reviews and Meta-Analyses (PRISMA) [13], as well as Murad and Wang's [14] guidelines for the composition of the manuscript.

### Search strategy and screening

One of us (DW), a systematic review librarian, systematically searched Embase and MEDLINE to locate systematic reviews limited to the management of diabetic neuropathy treatment (Fig 1). The librarian's searches were completed on June 2, 2020, with the search returns uploaded to Rayyan [15], a systematic review screening platform. After removing duplicates, in a masked, duplicate fashion, investigators (AK and HR) screened remaining articles by titles and abstracts to determine eligibility. Any discrepancies were noted and resolved by the two researchers through a consensus meeting.

### Eligibility criteria

For an article to be included in this study, the following must have been met: (1) the article must be a systematic review with or without a meta-analysis, (2) the review is focused on the management of diabetic neuropathy, (3) the article is retrievable in English, and (4) the article includes human participants. The Preferred Reporting Items for Systematic Reviews and Meta-Analysis Protocols (PRISMA-P) definition was used to define a systematic review [16].

| Ovid MEDLINE: | Ovid Embase: |
|---|---|
| 1. exp Diabetic Neuropathies/ | 1. exp diabetic neuropathy/ |
| 2. (diabet* adj2 (neuropath* or amyotroph* or polyneur* or mononeuropath* or neuralgia*)).mp. | 2. (diabet* adj2 (neuropath* or amyotroph* or polyneur* or mononeuropath* or neuralgia*)).mp. |
| 3. 1 or 2 | 3. 1 or 2 |
| 4. exp Therapeutics/ | 4. exp therapy/ |
| 5. (treat* or therap*).mp. | 5. (treat* or therap*).mp. |
| 6. 4 or 5 | 6. 4 or 5 |
| 7. exp "Systematic Review"/ | 7. exp "systematic review"/ |
| 8. exp Meta-Analysis/ | 8. exp meta analysis/ |
| 9. ("systematic review" or "meta-analysis" or (systematic* adj1 review*)).ti,ab. | 9. ("systematic review" or "meta-analysis" or (systematic* adj1 review*)).ti,ab. |
| 10. 7 or 8 or 9 | 10. 7 or 8 or 9 |
| 11. 3 and 6 and 10 | 11. 3 and 6 and 10 |

**Fig 1. The search strategy to obtain systematic reviews.**

## Training

An online training course for systematic reviews and meta-analyses by Li and Dickersin [17] was completed by investigators before screening commenced. The investigators also participated in four days of training focused on the definitions and identification of the nine most severe spin types that occur in systematic review abstracts as described by Yavchitz et al. [6] Lastly, investigators completed training over A MeaSurement Tool to Assess Systematic Reviews (AMSTAR-2) [18], an appraisal instrument used to determine the methodological quality of each included systematic review and meta-analysis. A comprehensive outline of author training can be found in our protocol published on Open Science Framework [12].

## Spin data extraction

The investigators (AK and HR) individually assessed every included systematic review for the presence of the nine most severe types of spin using a pilot-tested Google form for all data extraction—used previously by our team in another spin study [19]. The definition of these nine spin types can be found in Table 1. Following extraction, the two investigators (AK and HR) reviewed and discussed any inconsistencies within their results. When an agreement could not be reached, an additional author (RO) was available for adjudication.

## Data extraction of general characteristics

Additionally, investigators (AK and HR) extracted the following study characteristics from each systematic review: intervention type (pharmacological or non-pharmacological), whether a review discussed compliance with PRISMA [20], whether the associated journal required the compliance with PRISMA, the journal's 5-year impact factor, the study's funding source (not funded, industry, not mentioned, private, or public), and date the journals received the systematic reviews. Again, independent data extraction occurred with authors masked to each other's

**Table 1. Spin types and frequencies (%) in abstracts.**

| Nine most severe types of spin | No. (%), containing spin |
|---|---|
| 1) Conclusion contains recommendations for clinical practice not supported by the findings. | 0 (0) |
| 2) Title claims or suggests a beneficial effect of the experimental intervention not supported by the findings. | 0 (0) |
| 3) Selective reporting or overemphasis on efficacy outcomes or analysis favoring the beneficial effect of the experimental intervention. | 2 (1.8) |
| 4) Conclusion claims safety based on non-statistically significant results with a wide confidence interval. | 1 (1)* |
| 5) Conclusion claims the beneficial effect of the experimental treatment despite the high risk of bias in the included primary studies. | 5 (4.4) |
| 6) Selective reporting or overemphasis on harm outcomes or analysis favoring the safety of the experimental intervention. | 0 (0) |
| 7) Conclusion extrapolates the review's findings to a different intervention (i.e., claiming efficacy of one specific intervention although the review covers a class of several interventions). | 1 (0.9) |
| 8) Conclusion extrapolates the review's findings from a surrogate marker or a specific outcome to the global improvement of the disease. | 0 (0) |
| 9) Conclusion claims the beneficial effect of the experimental treatment despite reporting bias. | 3 (2.6) |

*9 did not assess safety, therefore n = 105.

responses. Following data extraction, authors were unmasked and discrepancies were resolved by group discussion.

## AMSTAR-2 rating

The methodological quality of each article was assessed via AMSTAR-2 (https://amstar.ca/), which rates systematic reviews from 'high' to 'critically low' based on a list of 16 criteria. The investigators (AK and HR) independently evaluated every included article utilizing the AMSTAR-2 tool, entering all data into a Google form. Articles that received different scores from the investigators were reexamined. All conflicts were resolved. The AMSTAR-2 has high reliability and validity compared to other systematic review appraisal tools, such as the original AMSTAR and Sach's Instrument [18].

## Statistical analysis

The study characteristics, spin frequency, and AMSTAR-2 ratings were reported as percentages and counts. In the protocol, we clarified the possibility of a binary logistic regression and assessed a power analysis to determine sample size from a previous study by our team [19]. Since the final sample size of 114 was not sufficiently powered to do multivariable logistic regression, we calculated unadjusted odds ratios which evaluated relationships between spin presence within abstracts and study characteristics. StataCorp 16.1 was used for the analyses [21].

## Results

### Characteristics

Our searches returned 1,297 articles, with 258 excluded as duplicates. During screening an additional 758 were excluded during abstract screening—578 because they were unrelated to diabetic neuropathy, 96 because they were ineligible by automation tools, 17 because they used animal subjects, and 5 because they were not related to diabetic neuropathy treatment. During the full-text review we excluded an additional 167 articles—68 due to wrong study design, 38 due to no relation to treatment, 18 due to the articles being only published abstracts, 16 due to being unavailable in English, 13 due to no full text being available, 7 due to articles only being protocols, 3 due to using animal studies, 3 due to articles being withdrawn, and 1 due to article being an erratum—resulting in a final sample size of 114 systematic reviews. The exclusion rationale and screening process are illustrated in Fig 2. The primary type of intervention was pharmacological therapies (77/114, 67.5%), followed by non-pharmacological interventions (37/114, 32.5%), and surgical interventions (0/114, 0%). The majority of the included studies did not mention adherence to PRISMA (81/114, 71.1%). Additionally, 74.6% of the journals did not recommend authors adhere to PRISMA guidelines (85/114). Of the 114 systematic reviews, 22 studies did not include a funding statement (21/114, 18.4%). Of the remaining, 77 received funding (67.5%), with public funding being the most common form of funding (44/114, 38.6%). Twenty-one studies did not include a funding statement (21/114, 18.4%), and 16 reported no funding (16/114, 14.04%). The dates the publishing journal received the systematic reviews ranged from 1993 to 2019 (Table 2).

### Spin in abstracts

Spin was identified in 9 of the 114 systematic review abstracts pertaining to diabetic neuropathy treatment (7.9%) (Table 1). However, certain abstracts contained multiple types of spin; thus, a total of 12 occurrences of spin were found. The most prevalent spin type was type 5 —"conclusion claims the beneficial effect of the experimental treatment despite the high risk of bias in the included primary studies"—which occurred in 5 abstracts (5/114, 4.4%). Spin

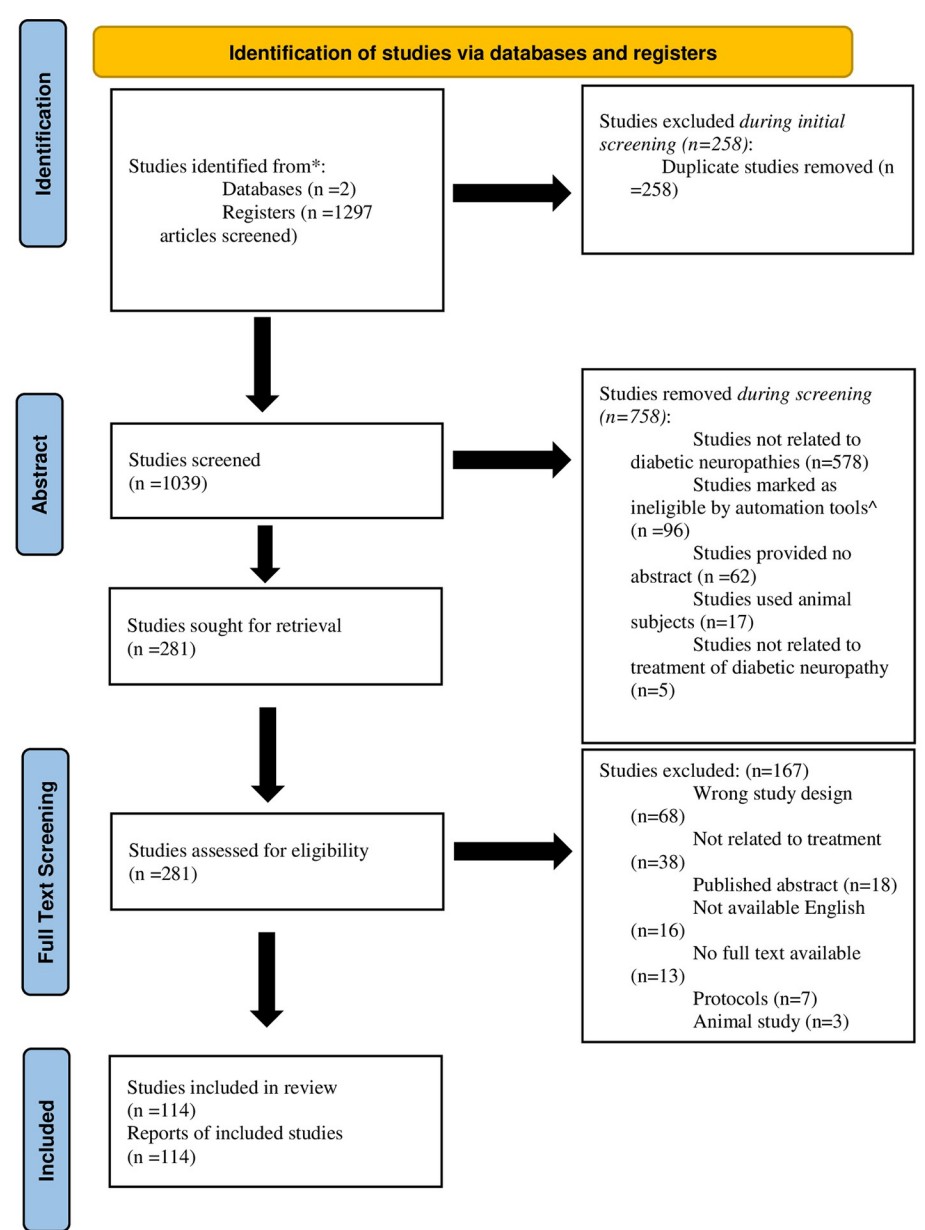

*Databases: Embase and MEDLINE

^Automation Tools: During the exclusion criteria, Rayyan had an automation feature that enabled automatic

exclusion criteria for any studies which didn't pertain to diabetic neuropathic treatment. These criteria would

**Fig 2. Flow diagram of study selection.**

**Table 2. General characteristics of systematic reviews and meta-analyses.**

| Characteristics | No. (%) of Articles (n = 114) | | | |
|---|---|---|---|---|
| | Total (%) | Abstract Without Spin | Abstract With Spin | Odds Ratio (95% CI) |
| Intervention type | | | | |
| Non-pharmacologic | 37 (32.5) | 33 (28.9) | 4 (3.5) | 1 [Ref] |
| Pharmacologic | 77 (67.5) | 72 (63.2) | 5 (4.4) | 0.57 (0.14–2.27) |
| Article mentions adherence to PRISMA | | | | |
| No | 81 (71.1) | 75 (65.8) | 6 (5.3) | 1 [Ref] |
| Yes | 33 (28.9) | 30 (26.3) | 3 (2.6) | 1.25 (0.29–5.32) |
| Publishing journal recommends adherence to PRISMA | | | | |
| No | 85 (74.6) | 77 (67.5) | 8 (7.02) | 1 [Ref] |
| Yes | 29 (25.4) | 28 (24.6) | 1 (0.9) | 0.34 (0.04–2.87) |
| Funding source | | | | |
| Not Funded | 16 (14.04) | 15 (13.2) | 1 (0.9) | 1 [Ref] |
| Industry | 22 (19.3) | 20 (17.5) | 2 (1.7) | 1.5 (0.12–18.13) |
| Not Mentioned | 21 (18.4) | 20 (17.5) | 1 (0.9) | 0.75 (0.04–12.99) |
| Private | 11 (9.6) | 10 (8.8) | 1 (0.9) | 1.5 (0.08–26.86) |
| Public | 44 (38.6) | 40 (35.1) | 4 (3.5) | 1.5 (0.15–14.52) |
| AMSTAR-2 Rating | | | | |
| High | 9 (7.9) | 8 (7.02) | 1 (0.9) | 1 [Ref] |
| Moderate | 31 (27.2) | 27 (23.7) | 4 (3.5) | 1.19 (0.12–12.17) |
| Low | 39 (34.2) | 37 (32.5) | 2 (1.7) | 0.43 (0.03–5.37) |
| Critically Low | 35 (30.7) | 33 (28.9) | 2 (1.7) | 0.48 (0.39–6.04) |
| Journal Impact Factor, M (SD) | 4.77 (4.37) | 4.93 (4.47) | 2.82 (2.10) | 0.75 (0.50–1.12) |

type 9—"conclusion claims the beneficial effect of the experimental treatment despite reporting bias"—was the second most frequent spin type, present in 3 abstracts (3/114, 2.6%). Nine studies did not mention safety measures in their abstract conclusion; therefore, spin type 4 was only observed in 1 of 105 studies (<1%). Spin types 1, 2, 6, and 8 did not occur in any abstracts. Our logistic regression models showed no significant associations between presence of spin within abstracts and a study's adherence to PRISMA, intervention type, funding source, nor if the publishing journal recommends PRISMA adherence or its 5-year impact factor (Table 2).

## AMSTAR-2 rating

AMSTAR-2 rated 7.9% of studies as 'high' quality (9/114), 27.2% as 'moderate' quality (31/114), 34.2% as 'low' quality (39/114), and 30.7% as 'critically low' quality (35/114) (Table 2). Approximately 99.1% of the systematic reviews developed their research question based on the Population, Intervention, Comparator group, Outcome (PICO) method (113/114) (Table 3). Additionally, 86.8% of authors explained the study selection design for inclusion (99/114). In contrast, 7% (7/114) of the systematic reviews used a comprehensive literature search strategy, and 14% (16/114) reported on the sources of funding for their analyzed reviews.

## Discussion

### General findings

Our study identified five of the nine most severe types of spin in systematic review abstracts involving diabetic neuropathy treatments. The most frequent type of spin was type 5: "conclusion claims the beneficial effect of the experimental treatment despite the high risk of bias in

**Table 3. AMSTAR-2 items and frequency of responses.**

| AMSTAR-2 Item | Response, n (%) | | |
|---|---|---|---|
| | Yes | No | Partial Yes |
| 1) Did the research questions and inclusion criteria for the review include the elements of PICO? | 113 (99.1) | 1 (0.9) | - |
| 2) Did the report of the review contain an explicit statement that the review methods were established prior to the conduct of the review and did the report justify any significant deviations from the protocol? | 51 (44.7) | 37 (32.5) | 26 (22.8) |
| 3) Did the review authors explain their selection of the study designs for inclusion in the review? | 99 (86.8) | 15(13.2) | - |
| 4) Did the review authors use a comprehensive literature search strategy? | 8 (7.02) | 27 (23.7) | 79 (69.3) |
| 5) Did the review authors perform study selection in duplicate? | 91 (79.8) | 23 (20.2) | - |
| 6) Did the review authors perform data extraction in duplicate? | 93 (81.6) | 21 (18.4) | - |
| 7) Did the review authors provide a list of excluded studies and justify the exclusions? | 30 (26.3) | 79 (69.3) | 5 (4.4) |
| 8) Did the review authors describe the included studies in adequate detail? | 56 (49.1) | 16 (14.04) | 42 (36.8) |
| 9) Did the review authors use a satisfactory technique for assessing the risk of bias (RoB) in individual studies that were included in the review? | 59 (51.8) | 34 (29.8) | 19 (16.7) |
| 10) Did the review authors report on the sources of funding for the studies included in the review? | 16 (14.04) | 98 (86.0) | - |
| 11) If meta-analysis was performed, did the review authors use appropriate methods for statistical combination of results?* | 64 (56.1) | 17 (14.9) | - |
| 12) If meta-analysis was performed, did the review authors assess the potential impact of RoB in individual studies on the results of the meta-analysis or other evidence synthesis?* | 58 (50.9) | 23 (20.2) | - |
| 13) Did the review authors account for RoB in primary studies when interpreting/ discussing the results of the review? | 78 (68.4) | 36 (31.6) | - |
| 14) Did the review authors provide a satisfactory explanation for, and discussion of, any heterogeneity observed in the results of the review? | 65 (57.02) | 49 (43.0) | - |
| 15) If they performed quantitative synthesis, did the review authors carry out an adequate investigation of publication bias (small study bias) and discuss its likely impact on the results of the review?* | 31 (27.2) | 50 (43.9) | - |
| 16) Did the review authors report any potential sources of conflict of interest, including any funding they received for conducting the review | 87 (76.3) | 27 (23.7) | - |

*33 systematic reviews did not perform a meta-analysis.

- Partial Yes was not applicable.

the included primary studies." One example of spin type 5 is from the abstract of a systematic review covering the efficacy of duloxetine in treating painful neuropathy, chronic pain, or fibromyalgia [22]. In the abstract, the authors claim a daily dose of 60 mg and 120 mg of duloxetine was an effective treatment for individuals with diabetic neuropathic pain. However, an examination of the article's full text revealed the majority of the primary studies included in this review had either a high risk of bias or the risk of bias was unclear. Omitting the risk of bias assessments from the abstract, as seen in this study, is problematic as studies with a high risk of bias can change the magnitude and direction of the study's outcomes–studies with a high risk of bias yield greater, and often incorrect, effect sizes as compared to studies with low-risk of bias [23,24]. As healthcare professionals often solely rely on the abstract [7], it becomes apparent a clinician's interpretation could be distorted. Therefore, we recommend authors of systematic reviews declare whether their included primary studies are at high risk of bias so clinicians can be cautious when interpreting the study's results.

Investigating spin in medical research has been of growing interest and has been well studied in randomized controlled trials. When comparing our results to other spin studies, spin occurs at a much lower frequency in systematic reviews focused on the management of diabetic neuropathy. The results of this study are encouraging as research over diabetic neuropathy seems to be at lower risk of containing spin compared to other fields. With these encouraging findings, the goal for diabetic neuropathy research should be focused on further eliminating the small amount of spin which occurs and preventing spin from occurring in future studies.

## Recommendations

Studies have shown misleading results in the abstract often go unrecognized by peer reviewers. For example, one study found reviewers failing to identify spin in nearly 80% of the studies reviewed [25]. In fact, this same study found 15% of reviewers asked the authors to include some form of spin in their abstract's conclusion [25]. Based on these findings, we first recommend spin training be provided to reviewers and editors as this would likely increase spin recognition and ultimately lead to its prevention. Additionally, we recommend the establishment of reporting guidelines which directly address spin and its subtypes as current guidelines fail to address spin. By having these guidelines, authors would be able to eliminate spin as it may be unintentionally included in the abstract. Lastly, we recommend clinicians using caution when interpreting the results of a study from only the abstract. Education covering spin may be beneficial to physicians and physicians in training.

## Strengths and limitations

Our study's strengths lie in its transparency, reproducibility, and measures to mitigate the effect of bias. All study materials were uploaded online prior to beginning our study, and any potential deviations from our protocol were recorded in an openly available update on OSF. Additionally, literature was reviewed in a double-masked fashion, the method deemed as the "gold standard" by the Cochrane Collaboration [26]. Limitations of our study include its cross-sectional nature, which reduces its generalizability, and the use of only MEDLINE and Embase to conduct our search, which may have lessened our sample size. Identifying spin is an inherently subjective process. However, to mitigate this subjectivity, investigators underwent intensive training to better characterize and identify spin.

## Conclusion

Overall, spin is infrequently observed in abstracts of systematic reviews covering diabetic neuropathy treatments. When comparing our results to other fields of medicine, systematic reviews related to diabetic neuropathy more often accurately portray the findings of the articles' full-text. Given that nearly 8% of these systematic reviews did contain spin, further reducing spin would lessen potential bias within these articles, which may be beneficial to other researchers, healthcare providers, and their patients.

## Author Contributions

**Conceptualization:** Matt Vassar.

**Data curation:** Micah Hartwell.

**Formal analysis:** Micah Hartwell.

**Funding acquisition:** Matt Vassar.

**Investigation:** Ali Khan, Haley Riley, Ryan Ottwell, Wade Arthur, Benjamin Greiner.

**Methodology:** Micah Hartwell, Matt Vassar.

**Project administration:** Micah Hartwell, Matt Vassar.

**Resources:** Ekaterina Shapiro, Stacy Chronister.

**Software:** Micah Hartwell.

**Supervision:** Ali Khan, Matt Vassar.

**Validation:** Micah Hartwell.

**Visualization:** Ali Khan, Haley Riley, Ryan Ottwell, Wade Arthur, Benjamin Greiner, Ekaterina Shapiro, Drew Wright, Micah Hartwell, Suhao Chen, Zhuqi Miao, Stacy Chronister.

**Writing – original draft:** Ali Khan, Haley Riley, Ryan Ottwell, Wade Arthur.

**Writing – review & editing:** Ali Khan, Haley Riley, Ryan Ottwell, Wade Arthur, Benjamin Greiner, Ekaterina Shapiro, Drew Wright, Micah Hartwell, Suhao Chen, Zhuqi Miao, Stacy Chronister, Matt Vassar.

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
