## [Decision Letter · Decision Letter 0]

28 Feb 2022

PONE-D-21-21584

The Presence of Spin in Systematic Reviews Focused on Diabetic Neuropathy: A Cross-Sectional Analysis

PLOS ONE

Dear Dr. Khan,

Thank you for submitting your manuscript to PLOS ONE. After careful consideration, we feel that it has merit but does not fully meet PLOS ONE’s publication criteria as it currently stands. Therefore, we invite you to submit a revised version of the manuscript that addresses the points raised during the review process.

A premise for acceptance is that the authors contributions become crystal clear.

The study protocol should be submited as addtional material. 

We look forward to receiving your revised manuscript.

Kind regards,

Tim Mathes

Academic Editor

PLOS ONE

https://journals.plos.org/plosone/s/fileid=ba62/PLOSOne_formatting_sample_title_authors_affiliations.pdf".

 “MV received the award.

The development of our protocol was funded by the Oklahoma State University Center for Health Sciences Presidential Mentor-Mentee Research Fellowship Grant.”

“The development of our protocol was funded by the Oklahoma State University Center for Health Sciences Presidential Mentor-Mentee Research Fellowship Grant.”

“MV received the award.

The development of our protocol was funded by the Oklahoma State University Center for Health Sciences Presidential Mentor-Mentee Research Fellowship Grant.”

Reviewers' comments:

Reviewer's Responses to Questions

**Comments to the Author**

1. Is the manuscript technically sound, and do the data support the conclusions?

Reviewer #1: Yes

Reviewer #2: Yes

2. Has the statistical analysis been performed appropriately and rigorously? 

Reviewer #1: Yes

Reviewer #2: Yes

3. Have the authors made all data underlying the findings in their manuscript fully available?

Reviewer #1: No

Reviewer #2: No

4. Is the manuscript presented in an intelligible fashion and written in standard English?

Reviewer #1: No

Reviewer #2: Yes

5. Review Comments to the Author

Reviewer #1: Khan and colleagues performed a systematic review to evaluate the presence of nine of the most severe types of spin (as identified by Yachitz et al.) in the abstracts of systematic reviews on diabetic neuropathy. This work builds upon a number of similar evaluations of spin of a variety of disciplines by their group, and therefore the results can be compared to other research topics, which they have done to an extent in the discussion. The methods appear to be rigorously executed from what is provided, although not all materials are able to be evaluated by the reviewer to make a final determination (see first comments).

Major comments

"All relevant data are within the manuscript and/or Supporting Information files"

- Unless I missed it, I do not see raw data available to review.

Line 86: "To ensure our study’s reproducibility and transparency, this study’s protocol, extraction forms, and data analysis scripts can be found in Open Science Framework (OSF) [link]."

- This link is not active (permission error) for the reviewer to check these items. Please allow public access or attach the items for review if there is a reason for these items not to be public before publication.

PRISMA flow diagram

- Please provide enough information so a reader can reproduce your process of using automation tools that were used to screen records, and so readers can assess the reliability of doing this.

Do the following statements contradict one another?

- "Since the final sample size of 114 was not sufficiently powered to do multivariable logistic regression, we calculated unadjusted odds ratios which evaluated relationships between spin presence within abstracts and study characteristics."

- "Our logistic regression model found no significant association between abstracts with spin and a study’s PRISMA adherence, intervention type, if the publishing journal recommends PRISMA adherence, funding source, or the 5-year impact factor of the publishing journal (Table 2)."

Minor comments

Abstract

- Conclusion: I agree that compared to other fields, review abstracts on diabetic neuropathy seem to contain less spin. Yet, 8% of studies still seems unacceptable, and therefore I would not use the language that they "accurately portray the findings of the review's full-text". Your conclusion (line 211) seems to agree that this is encouraging but we should not be complacent.

Line 69: "physicians must stay updated through research and various learning platforms".

- Do you mean by reading the scientific literature?

70: "Comprehensive study analyses which comprise multiple studies" -> change to "Systematic reviews and meta-analyses", or rephrase another way.

Language

- Please edit the manuscript carefully for grammar and spelling, including the abstract.

Reviewer #2: Thank you for the opportunity to review the manuscript entitled “The Presence of Spin in Systematic Reviews Focused on Diabetic Neuropathy: A Cross-Sectional Analysis”.

The authors have performed a meta-research study that consisted of a systematic search for systematic reviews on diabetic neuropathy treatment and an analysis of the prevalence of different types of spin. They also extracted review characteristics and assessed the included review’s methodological quality using AMSTAR 2, and then evaluated whether there was an association between these characteristics or quality and the prevalence of spin.

While this is an important field of study and valuable research question, there are several aspects concerning the manuscript that should be addressed:

Major:

1. Apart from AK, HR, DW and MV, it is currently unclear how the co-authors were involved in the study process. Please provide an author statement using the CRediT Taxonomy.

2. Introduction: Argumentation:

a. I cannot entirely follow the argumentation around systematic reviews, e.g. “Because systematic reviews are the gold standard it is important that they are conducted without bias or misleading findings”. I would consider it the other way around: By definition, systematic reviews use explicit, systematic methods to reduce the risk of bias in the review process (bias cannot be eliminated), therefore they are considered the highest level of evidence (for most questions). Given this high standing, they are supposed report findings in a way that is not misleading. However, while spin has long been known to occur in RCTs, there is now evidence for spin in systematic reviews, too (citations of such studies).

b. The concept of spin should be elaborated more in the argumentation, e.g. introduce the classification by Yachitz before the aims & objective statement.

c. The point about physicians often only reading abstracts should also be moved up to the argumentation, i.e. before the aims & objective statement.

3. Introduction: Choice of references: It is unclear why spin in anesthesiology and emergency medicine RCTs is mentioned here; I would suggest citing studies of spin in RCTs on diabetic neuropathy treatment, if available, or an evaluation of spin in RCTs that is not focused on a specific field. If there is a reason to believe that the medical field influences the amount of spin, it would be interesting to cite such evidence in the introduction. Regarding spin in abstracts of systematic reviews: it would be interesting to present a brief overview of the findings of the other meta-research studies conducted by the team around MV (at least 18 of which have already been published according to a PubMed search).

4. Methods: Protocol and Reporting: The files in OSF do not appear to be public (when I tried it (as a logged-in OSF user), it said “You Need Permission”). Please make your files public, otherwise the reproducibility and transparency of your study (which you name as a strength) is not given.

5. Methods: Protocol and Reporting: Please provide more information on the third-party analytical team.

6. Methods: Protocol and Reporting: While I greatly encourage the use of reporting guidelines, it is unclear to me why PRISMA 2020 was used for this meta-research study. It also appears to me that there were some misunderstandings, i.e., in the PRISMA checklist it says “Yes/Table 3” for the items 13e (Describe any methods used to explore possible causes of heterogeneity among study results (e.g. subgroup analysis, meta-regression)) and 14, neither of which was applicable in this study. Furthermore, line numbers need to be updated if use of PRISMA 2020 is continued (or changed to page numbers may be more feasible).

7. Methods: Data Extraction: The column in Table 1 that contains results must be reported in the results section please. The remaining table would benefit from examples and should contain the citation of Yachitz et al.

8. Methods: Data Extraction: The description of AMSTAR-2 assessment should be an extra paragraph containing an explanation of how the overall assessment was derived at. Furthermore, the tool needs to be cited correctly (https://www.bmj.com/content/358/bmj.j4008) and it is currently unclear what you mean by “high construct validity coefficients based on the original AMSTAR-2 instrument (r = 0.91) and the (r = 0.84).[16]”.

9. Methods: Data Extraction: Please provide the categorization system for each characteristic and add that you also extracted the date the journals received the systematic reviews.

10. Results: Characteristics: Please provide a list of included reviews as well as a list of exclusions with reasons.

11. Results: AMSTAR-2 Rating: Please provide the results per individual review in a supplemental file.

12. Discussion: Please discuss potential reasons why systematic reviews focused on the management of diabetic neuropathy tend to have less spin than systematic reviews from other fields.

13. Discussion: It is unclear why Ottwell et al. and Heigle et al. are being cited. It would be more informative to report the range of spin across the systematic review abstracts of various clinical disciplines that the team around MV have investigated.

Minor:

14. To better differentiate between studies as in other meta-research studies and the included articles, I would suggest to re-word them to “included reviews”, “analysed reviews”, etc.

15. Abstract: Methods: “…, of which included 114 systematic reviews for spin assessment.” Suggest to insert “we” in front of “included”.

16. Abstract: Methods: Please state that you mean the nine most severe spin forms as defined by Yachitz et al. 2016.

17. Abstract: Methods: Please include a brief description of the analyses performed.

18. Abstract: Results: “No association exists”, please reword to “was observed”.

19. Abstract: Conclusions: Please change to “whose abstracts mostly portray the findings of the review’s full-text accurately.” to reflect the results adequately.

20. Methods: Search Strategy: Please rename section to “search strategy and screening” and describe if there were any limits to the search and how you used the automation function of Rayyan.

21. Methods: Fig 2 is mentioned before Fig 1, please correct.

22. Results: Figure 1: It should “diagram” not “diaphragm”. Please provide a legend for the asterisks and correct the number of records identified from databases (= MEDLINE and Embase) to 1297 and the number of records identified from registers to 0.

23. Results: Characteristics: “The primary type of intervention was pharmacological therapies (77/114, 67.5%), followed by non-pharmacological interventions (37/114, 32.5%), and nonsurgical interventions (0/114, 0%).” Do you mean “surgical interventions”? Otherwise, the categories are not mutually exclusive.

24. Results: Characteristics: “77 received funding (67.5%)” this value is not completely correct, as 21 did not mention funding but may have been funded. Suggest to re-organize to: “Twenty-one studies did not include a funding statement (21/114, 18.4%). Of the remaining, 77 received funding (67.5%), with public funding …”

25. Results: Table 2: Journal Impact Factor, M (SD): What is the value in the fourth column? If it is an odds ratio, what was the reference?

26. Results: Table 3: In the right column, please indicate the items where partial yes was not applicable.

27. Conclusion: Please change to “whose abstracts mostly portray the findings of the review’s full-text accurately.” to reflect the results adequately.

6. PLOS authors have the option to publish the peer review history of their article (what does this mean?). If published, this will include your full peer review and any attached files.

Reviewer #1: **Yes: **Colby Vorland

Reviewer #2: **Yes: **Tanja Rombey

---

## [Author Response · Author response to Decision Letter 0]

22 Apr 2022

All comments are provided in our documentation titled “Response to Reviewers” which is attached to the manuscript submission. We are truly grateful for all the feedback received from the reviewers, as it truly allowed our manuscript to improve.

---

## [Decision Letter · Decision Letter 1]

23 May 2022

PONE-D-21-21584R1The presence of spin in systematic reviews focused on diabetic neuropathy: a cross-sectional analysisPLOS ONE

Dear Dr. Khan,

Thank you for submitting your manuscript to PLOS ONE. After careful consideration, we feel that it has merit but does not fully meet PLOS ONE’s publication criteria as it currently stands. Therefore, we invite you to submit a revised version of the manuscript that addresses the points raised during the review process.

We look forward to receiving your revised manuscript.

Kind regards,

Tim Mathes

Academic Editor

PLOS ONE

Journal Requirements:

Reviewers' comments:

Reviewer's Responses to Questions

**Comments to the Author**

1. If the authors have adequately addressed your comments raised in a previous round of review and you feel that this manuscript is now acceptable for publication, you may indicate that here to bypass the “Comments to the Author” section, enter your conflict of interest statement in the “Confidential to Editor” section, and submit your "Accept" recommendation.

Reviewer #1: (No Response)

Reviewer #2: (No Response)

2. Is the manuscript technically sound, and do the data support the conclusions?

Reviewer #1: Partly

Reviewer #2: Yes

3. Has the statistical analysis been performed appropriately and rigorously? 

Reviewer #1: I Don't Know

Reviewer #2: I Don't Know

4. Have the authors made all data underlying the findings in their manuscript fully available?

Reviewer #1: No

Reviewer #2: No

5. Is the manuscript presented in an intelligible fashion and written in standard English?

Reviewer #1: Yes

Reviewer #2: Yes

6. Review Comments to the Author

Reviewer #1: I thank the authors for their replies to my comments. My remaining comments are below.

PRISMA flow diagram

Original comment: Please provide enough information so a reader can reproduce your process of using automation tools

that were used to screen records, and so readers can assess the reliability of doing this.

Author response: The PRISMA flow diagram is updated to where the reader will be able to reproduce the process of

screening records.

New comment: There is still a vague reference to automation in the flow diagram, and a new detail in the methods that "96 because they were ineligible by automation tools". Which automation tools? How did they determine ineligibility? Why were they ineligible? How did you determine you could trust the algorithms?

Thank you for providing access to your protocol, and for its comprehensiveness for reproducibility.

In the protocol, "To promote the reproducibility of our study, the study protocol, raw data, analysis scripts, data dictionaries, training videos, and extraction forms will be deposited on Open Science Framework." Do you plan to publish analysis scripts and data dictionaries in the repository?

Thank you for attaching your data. It is not clear whether the editorial software converted the data files to PDF form, but it is extremely difficult to review these as PDFs. For instance, there are different colors in cells, it is not clear if there is a data dictionary describing what these colors mean, and I cannot read most of the text in the cells. Please include them as spreadsheets either as supplemental files or in your OSF repository.

The current data availability statement "All relevant data are within the manuscript" is not clear; if the data will be supplementary files or in the repository, this can be stated.

Reviewer #2: The authors have satisfactorily addressed most of my previous comments. However, some important comments need further attention before their manuscript is accepted for publication.

1. Unfortunately I could not find the CREDIT Statement of authors' contributions. Furthermore, in the marked version of the manuscript, it appears to me that the order of authors has been changed since the initial submission. Please ensure that authorship is consistent in both manuscript versions (marked and unmarked). If the order of authors has indeed changed, this should also be reflected in the CREDIT statement (i.e. did they perform major work in the revision?).

2. Data sharing/transparency: The OSF page is now openly accessible, which is good. I was also pleased to see that you submitted the list of in- and excluded studies, details of included studies and spin ratings, and AMSTAR ratings as supplemental files. However, the current format (Excel sheets converted to PDF) is not fully readable. Without a legend it is also unclear what the colours mean. My understanding is that this is not the authors' fault, but the submission system (editorial manager) converts all document types to PDF. I therefore suggest that you convert the files to Word documents with legends and ensure every cell is fully readable before submitting them. Additionally (or instead) you may want to upload your Excel files to OSF.

3. In the abstract's conclusion there was a misunderstanding about my comment. I did not mean "to reflect the results adequately”. to be part of the sentence. However, I leave it up to the authors if they want to change it.

4. PRIMSA compliance as a subheading appears to be misleading. Suggest to move this section after "Spin data extraction" and name it "Data extraction of general characteristics" or similarly.

5. The odds ratio for JIF is still unclear to me; did you take the mean JIF and calculated the odds for journals below and above the mean JIF? If yes, please insert two rows (as for the other characteristics) and indicate which is the reference row.

7. PLOS authors have the option to publish the peer review history of their article (what does this mean?). If published, this will include your full peer review and any attached files.

Reviewer #1: **Yes: **Colby Vorland

Reviewer #2: **Yes: **Tanja Rombey

---

## [Author Response · Author response to Decision Letter 1]

15 Aug 2022

All of the responses for each of the reviewer and editor comments are viewable in the Second Rebuttal Letter. We thank the reviewers for allowing us the opportunity to improve our manuscript.

---

## [Decision Letter · Decision Letter 2]

6 Sep 2022

The Presence of Spin in Systematic Reviews Focused on Diabetic Neuropathy: A Cross-Sectional Analysis

PONE-D-21-21584R2

Dear Dr. Khan,

We’re pleased to inform you that your manuscript has been judged scientifically suitable for publication and will be formally accepted for publication once it meets all outstanding technical requirements.

Kind regards,

Tim Mathes

Academic Editor

PLOS ONE

Additional Editor Comments (optional):

Reviewers' comments:

Reviewer's Responses to Questions

**Comments to the Author**

1. If the authors have adequately addressed your comments raised in a previous round of review and you feel that this manuscript is now acceptable for publication, you may indicate that here to bypass the “Comments to the Author” section, enter your conflict of interest statement in the “Confidential to Editor” section, and submit your "Accept" recommendation.

Reviewer #1: All comments have been addressed

Reviewer #2: All comments have been addressed

2. Is the manuscript technically sound, and do the data support the conclusions?

Reviewer #1: Yes

Reviewer #2: Yes

3. Has the statistical analysis been performed appropriately and rigorously? 

Reviewer #1: Yes

Reviewer #2: I Don't Know

4. Have the authors made all data underlying the findings in their manuscript fully available?

Reviewer #1: Yes

Reviewer #2: Yes

5. Is the manuscript presented in an intelligible fashion and written in standard English?

Reviewer #1: Yes

Reviewer #2: Yes

6. Review Comments to the Author

Reviewer #1: (No Response)

Reviewer #2: Thank you for addressing all my previous comments!

I do not have any further comments other than I believe it would be beneficial for the reader if the explanation for the interpretation of the OR for the JIF would be included in a legend below the table. However, I leave this decision to the authors.

7. PLOS authors have the option to publish the peer review history of their article (what does this mean?). If published, this will include your full peer review and any attached files.

Reviewer #1: **Yes: **Colby Vorland

Reviewer #2: **Yes: **Tanja Rombey

---

## [Editor Report · Acceptance letter]

11 Sep 2022

PONE-D-21-21584R2 

The Presence of Spin in Systematic Reviews Focused on Diabetic Neuropathy: A Cross-Sectional Analysis 

Dear Dr. Khan:

I'm pleased to inform you that your manuscript has been deemed suitable for publication in PLOS ONE. Congratulations! Your manuscript is now with our production department. 

Kind regards, 

on behalf of

Dr. Tim Mathes 

Academic Editor

PLOS ONE